# Numerical Study of Water-Oil Two-Phase Flow Evolution in a Y-Junction Horizontal Pipeline

**M. De la Cruz-Ávila** [1,*] **, I. Carvajal-Mariscal** [2] **, Leonardo Di G. Sigalotti** [3] **and Jaime Klapp** [1,*]

1 Instituto Nacional de Investigaciones Nucleares (ININ), Carretera México Toluca km. 36.5, La Marquesa, Ocoyoacac 52750, Mexico
2 Instituto Politécnico Nacional, ESIME-UPALM, Mexico City 07738, Mexico
3 Departamento de Ciencias Básicas, Universidad Autónoma Metropolitana-Azcapotzalco, Av. San Pablo 180, Mexico City 02200, Mexico
* Correspondence: mauriciodlca1@gmail.com (M.D.l.C.-Á.); jaime.klapp@inin.gob.mx (J.K.); Tel.: +52-(55)-53297200 (J.K.)

**Abstract:** The work aims to numerically evaluate different injection configurations for the analysis of a two-phase flow behavior and evolution through a staggered Y-junction pipeline. To minimize agglomeration between inlets, the injection zones have a separation distance, avoiding areas with eddies or swirls owing to strong turbulence. Six input scenarios were examined accordingly with injection system experimental data. Results show significant variations because the main fluid develops a swirl over the pipe center. This is generated immediately after the phases' supply zone due to the oil-phase because it presents a partial pipe flooding, even in the water injection zone. Moreover, the supply configuration has significant relevance to the main flow development. Accordingly, many flow patterns can be achieved depending on the phases' confluence coming from the supply system. The interface velocities confirm the transition process and flow pattern development, which are driven by the phases' velocities describing the early stages of three flow patterns formed during the fluids' confluence. Finally, a substantial extent of the conjunction process points out that caution must be exercised during the injection supply system selection for this type of junction pipeline to achieve a better, and smooth blend, with either narrow, medium, or wide emulsions.

**Keywords:** numerical study; Y-junction pipe; two-phase flow; high-viscosity fluids; flow evolution; water-oil flow

## 1. Introduction

The two-phase oil-water flow is prevalent in a variety of industries, including biochemical [1], energy, and petroleum extraction, transportation, and storage [2]. Because of the turbulent momentum and interfacial tension [3] that effect each phase, oil and water typically flow in the dispersed phase and continuous phase, respectively [3]. Significant variations in density and kinematic viscosity among both, oil and water, combined with changes of the phase velocity, phase fraction, and pipe geometry, will influence flow conditions, resulting in complicated and variable flow patterns [4]. Furthermore, the slip velocity exists across the two phases [5]. As a result of the complicated flow structure and slip phenomena, properly measuring the flow velocity is challenging [6,7].

Diverse researchers [6,8–11] have investigated oil and water flows extensively with the purpose of characterizing the resultant flow patterns, pressure gradients, in situ phase fractions, and interfacial phenomena under a range of different flow regimes, fluid characteristics, and pipe designs. Unfortunately, the nature of these flows is still not totally understood because to the complicated flow dynamics seen in pipelines that involve more than liquid-liquid flows where the densities and viscosities of the two fluids are significantly different. Due to their impact on flow pattern transitions [6] and interface curvature [12], additional parameters such as interfacial forces, wetting properties [13], phase holdup [14–16],

phase inversion [4], and the most interesting, miscible characteristic are important factors to consider [17] in pipe junctions flows.

The T-junction [18–20] and Y-junction [16,21] are two popular pipeline supply methods in industrial, chemical, and petroleum engineering [22,23]. There are many studies of junction flow, including oil-water two-phase flow [24], thermal mixing [25], and turbulent mixing [26], to which researchers are focusing due to variation rules in the various mixing phases. Many studies have compared the properties of the Y-junction with the T-junction, for example, Sierens and Verhelst [27] investigated the effect of injection parameters on four types of junctions, T-junction, Y-junction, 45-degree junction, and 45-degree reverse junction, and found that the Y-junction produced the highest power and 45-degree junction produced the highest efficiency.

The turbulent flow in junction mixing is another significant research area. In these earlier works, computational fluid dynamics (CFD) is a common non-intrusive method for measuring properties and describing behaviours using numerical models [28]. The $k - \varepsilon$ model [29] and $k - \omega$ SST model [30,31] have been used to describe the turbulence in numerical simulations. The SST model which is based on the $k - \omega$, is used to model the boundary layer outside, which results are more accurate.

### 1.1. Aim and Scope

Knowing how the phases' conjunction develops from the supply system, substantial improvements can be made to enhance the oil's transport in pipes, as well as to improve the techniques already used. As a result, the goal of this research is to use numerical simulations to examine fluid-flow development in a two-phase system with an equal proportion of water and oil, but distinct supply entry, minor fouling, and clogging in a Y-junction round horizontal pipe. Moreover, it can help to better choose in advance the resolution of the holdup correlation type based on the flow pattern. Last but not least, it can help to better propose the numerical solution schemes not only based on the multiphase scheme, whether "non-miscible" or "mixed" but with the consideration of selecting them based on the phase interpenetration.

One of the main purposes of this investigation is to confirm that the emulsification process is not only driven by the interface or slip velocities at an early stage of the confluence supply entrances but also by the two-phase flow pattern. The second is to verify that the flow pattern, i.e., early stage of emulsions, develops according to the phases' velocities during the fluids' confluence. Finally, to reveal if the supply configuration system has significant relevance to the development of the main fluid flow and the substantial extent of an emulsification process.

### 1.2. Implementation

In this study, we present the results obtained numerically using the RANS technique coupled with the Volume of Fluid (VOF) method, the appliance of the $k - \omega$ SST turbulence model and high interface reconstruction scheme which have taken on greater relevance and precision of results in the Computational Fluid Dynamics (CFD) study models for two-phase systems. Through the use of the VOF multiphase model, the phases are modelled for the representation of a flow in two phases of water-oil. It was possible to obtain the interfacial or slip velocity, as well as the volumetric fraction of the phases during the conjunction, showing that the fluid development process in general directly influences the "mixing" of phases and the development of flow patterns.

When volume fraction or phase fraction is used in immiscible fluids, layers of fluid are presented rather than bubble-like or drop-like structures, since most post-processing programs assign a cut-off function or condition to those layers. If care is not taken in advance to indicate which structures within the bulk fluid are considered as part of the confluence interaction, it might not actually exist, which is why they are assumed as misinterpretations. Therefore, an analysis of the strain rate must be made together with the shear velocity [17], even when these structures exist in the form of layers, it is analyzed how the bulk-fluid is deformed since the development of the flow pattern is not yet constituted.

With these results, an addition can be made to the numerical multiphase models for the correct selection of some coupled model for mass transfer based on diffusion and slip velocity, as well as the flow pattern based on specific dimensionless numbers such as the Sherwood, *Sh*, Schmith, *Sc*, and multiphase Reynolds number, *Re*$_{qp}$. With this information it is also possible to select the correct correlation for the calculation of the holdup of each phase.

In what follows, Section 2 briefly describes the experimental testing apparatus, which specific details are presented in the respective citation while Section 3 exposes each and every one of the particularities of the numerical methodology used. Section 4 presents the analysis of the results and the discussion of the findings obtained are carried out and Section 5 summarizes the relevant conclusions.

## 2. Experimental Setup

Figure 1 depicts the experimental setup pipeline that is located at the Engineering Institute-Universidad Nacional Autónoma de México (UNAM). This facility was designed to evaluate the flow properties of liquid-liquid and liquid-air mixtures constituted by high-viscosity liquids. The **P1** sensor data is used for this analysis where experimental rig is detailed described in [32–34]. For these particularly numerical simulations, the geometry or computational domain were discretized explicitly on the Y-junction supply system to focus computational resources under the evolution of main fluid development. The confluence system was selected and marked as Injection Point illustrated in the close-up Figure 1B. Further details are described in [32].

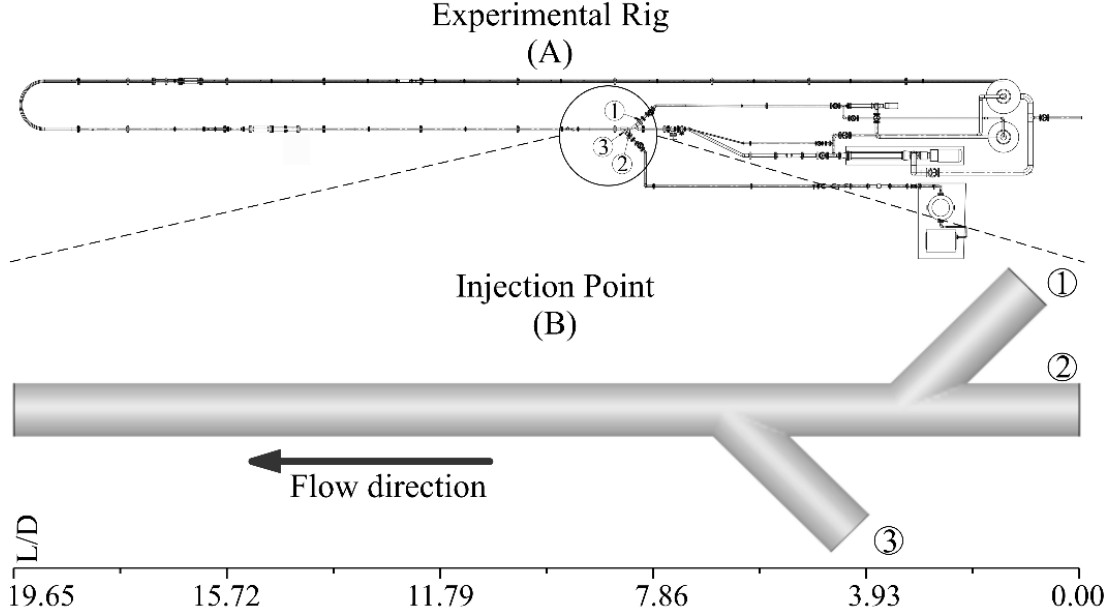

**Figure 1.** Top view of the complete experimental rig setup (**A**), injection point close-up (**B**). (1), (2) and (3) Phases Y-junction injection system. L is the characteristic length and D the characteristic diameter of the pipe respectively for the dimensionless representative scale.

## 3. Numerical Setup

### 3.1. Case Study

A staggered Y-junction nozzles with a distance of 25 cm between them and a 45° angled junction was selected for the numerical domain model. All injectors have a pipe diameter, D, of 7.62 cm and a total length, L, of 150 cm. ANSYS Fluent CFD commercial software [35] in a Xeon 32 cores Workstation and two high-performance GPUs, Nvidia Quadro 6000 and a Tesla C2075 to accelerate calculation was used to run two-phase water-oil 3D simulations. For both fluids, the numerical domain is shown as a horizontal Y-junction pipe with two confluence zones to the main pipeline. According to the research supply configurations [17], the input mass flow is constant at 5 kg/s for both phases and is distributed symmetrically

across the injection pipes. The continuous phase is marked as pure water fluid with the corresponding thermophysical properties and the second phase or dispersed phase is marked as oil fluid.

It is noteworthy that the second fluid, labelled as oil, has thermophysical properties set as the same as those of the fluid used in the experiments, which are displayed elsewhere [34,36]. Glycerol is a very viscous and hygroscopic liquid; it absorbs moisture from the air. Due to the fact that it is miscible with water in all proportions, it never becomes "insoluble" in water. This result in, to determine the general physical emulsification process, it was not take into account the chemical dynamics [37,38] and the hydrogen bonding velocity [39] during the conjunction. Therefore, even though the second phase has the same thermophysical properties as glycerol, it was treated as an immiscible fluid. All this with the main objective of focusing on the effects of its viscous and density properties in the merely physical process of the bulk fluid evolution and development during the first stages of supply into the pipe. This characterization of the phases allows for the application of the VOF model.

To minimize agglomeration or collision between the inlets, the injection zones have a separation distance, with the goal of reducing excessive turbulence and therefore avoiding regions with eddies or swirls over the main channel. Six input cases or combinations were examined using the phase procedure. Three cases for the incorporation of the low viscosity fluid through two alternate supply zones and three other cases for the high viscosity fluid respectively. The numerical simulations test matrix is listed in Table 1.

**Table 1.** Phases injection input scenarios.

| Case Studies | 1 | | | 2 | | | 3 | | |
|---|---|---|---|---|---|---|---|---|---|
| Nozzle | 1 | 2 | 3 | 1 | 2 | 3 | 1 | 2 | 3 |
| Water | X | O | O | O | X | O | O | O | X |
| Glycerol | O | X | X | X | O | X | X | X | O |
| **Case Studies** | **4** | | | **5** | | | **6** | | |
| Nozzle | 1 | 2 | 3 | 1 | 2 | 3 | 1 | 2 | 3 |
| Water | O | X | X | X | O | X | X | X | O |
| Glycerol | X | O | O | O | X | O | O | O | X |

*3.2. Numerical Domain Details*

The numerical domain mesh was created using a combination of advancing-front meshing [40] and the Cut-Cell technique [41]. The advancing-front technique offers several advantages over traditional grids, including the ability to alter the mesh density and facilitate tessellation in geometrically complicated domains. The staggered nozzles, which the Cut-Cell mesh must be fitted to, are the reason of the geometry's refinement, particularly at the Injection Point, as seen in Figure 2. When these two methods are used, a steady increase of thin layers and cells from the pipe walls to the domain's border is obtained, allowing a good approximation of the physical phenomenon, especially when high-order discretization schemes are applied.

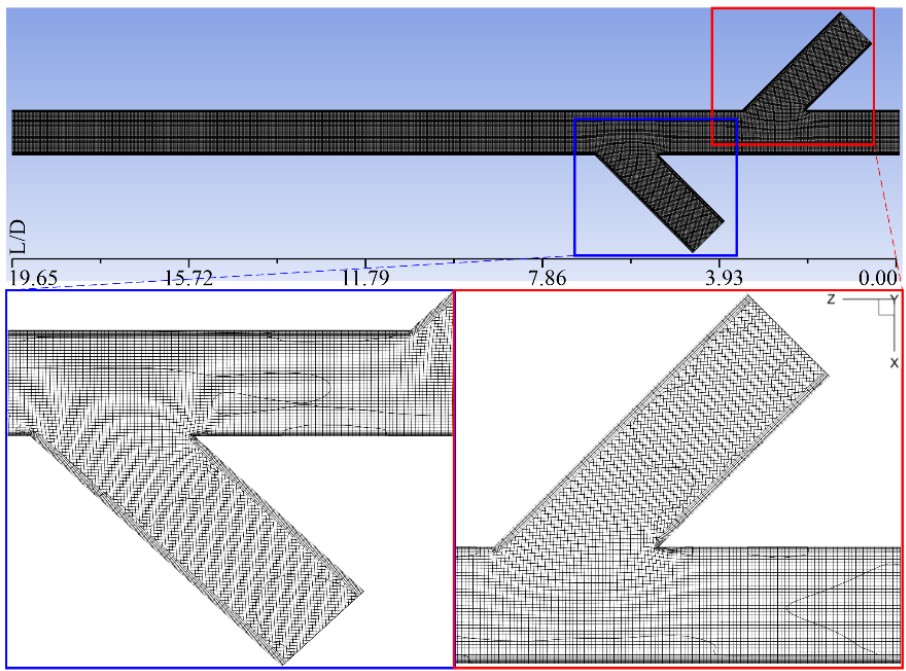

**Figure 2.** Cut-Cell hexahedral mesh for the phase injection system.

In the simulations, adiabatic and non-slip enhanced wall function to correct any miscalculation near walls treatment were employed as boundary conditions. Because a desire Y+ in these types of two-phase simulations, which are mostly driven by the high-viscous flow, is from 1.5 to 2, a generalized Y+ was taken mostly desirable as 1 for the initial-cell-length of the constructed meshes whose outcome was to get the correct length of the first cell at 1.6. This Y+ was taken into account also for the fastest phase, which in this case was water as the transport phase. The first grid length on the walls is 0.000235 m with an increment of 20% to complete 10 layers. In each nozzle, a mass flow condition was employed to inject the phases. The discharge occurs at the outlet limit at atmospheric conditions of 1 atm of pressure and 293 K of temperature.

### 3.3. Numerical Domain Details

The numerical simulations were discretised as follows. For pressure spatial discretization the PRESTO Scheme (PREssure-STaggering-Option) [42], a third order MUSCL (Monotonic Upstream-centred Scheme for Conservation Laws) [43] for momentum solution and the modified HRIC (High Resolution Interface Capturing) [44,45] for the volume fraction reconstruction. A completely linked approach was also built for the pressure-velocity solution technique. The results demonstrate that, as compared to first-order techniques, the high-order MUSCL method efficiently minimizes numerical diffusion, resulting in superior two-phase flow resolution. Furthermore, when compared to other options, the PRESTO and HRIC with the pressure-velocity fully coupled operating ensemble demonstrate greater convergence.

In addition, in this kind of two-phase simulation, an adaptable step size was included for the numerical simulations to guarantee that the time-step was appropriate for the fluid flow development. The interval of time should be short enough to solve time-dependent features and provide convergence in the timeframe given. The time step is calculated using equation,

$$\Delta t = \frac{Typical\ Cell\ Size}{Characteristic\ Flow\ Velocity},\qquad(1)$$

and the values considered in the simulations have the order of $2.25 \times 10^{-5} < \Delta t < 5 \times 10^{-3}$.

The most suited two-phase model for monitoring the surface of the two immiscible fluids is the so-called Volume of Fluid (VOF, or surface-tracking approach), owing to the hydrodynamics of the water-oil flow in this work.

The standing features of the VOF technique are:

- Accuracy in the prediction of two-phase flow. In this case, the high viscous water-oil two-phase because it considers non-interpenetrating phases;
- Utilization of Third-order discretization schemes for phase tracking;
- The high phase reconstruction schemes for the volume fraction;
- More flexibility and efficiency than the finite-difference method. This includes dealing with issues requiring extremely complex free surface configurations;
- It provides a simple and affordable way to monitor phases on three-dimensional grids;
- It runs in high-performance GPUs parallel processing.

In this technique, the phases are considered as continuous, which prevents them from interpenetrating. Additionally, the phases in this approach are isothermal, transitory, and do not include mass transfer or phase change. As a result, each of the fluids under consideration has a single set of momentum equations, and the volume fraction of each fluid in each computing cell is traced across the domain.

When utilizing the VOF model, there are a few things to keep in mind ahead of time to ensure a good numerical description. That is, the volume fractions of all phases in each control volume must amount to one. The fields for all variables and attributes are shared by the phases and represent volume-averaged values as long as the volume fraction of either of the phases is known per each location. As a result, the variables and features in any particular cell are either solely reflective of one of the *th*-phases, or indicative of the phases' combination, depending on the volume fraction values,

$$\alpha_q = \left\{ \begin{array}{l} 0 \rightarrow Cell\ is\ empty\ of\ the\ q^{th}\ fluid \\ 1 \rightarrow Cell\ is\ full\ of\ the\ q^{th}\ fluid \end{array} \right\} \tag{2}$$

$$0 < \alpha_q < 1 \;\rightarrow\; The\ cell\ contains\ the\ interface. \tag{3}$$

Based on the local value of the phase $\alpha_q$ the appropriate properties and variables will be assigned to each control volume within the domain. In this study, the primary phase is water, $\alpha_w$.

*3.4. Governing Equations*

The solution of the continuity equation for the phase volume fraction allows the tracking of the interface between the phases, and is given by:

$$\frac{\partial \rho}{\partial t} + \nabla \cdot \rho \vec{v} = \sum_m S_n, \text{z} \tag{4}$$

where $\rho$ is the density, $\vec{v}$ the velocity vector, $t$ the time and $S = 0$ due to the no mass transfer assumption. For the interfacial tracking, oil as the secondary phase, $\alpha_g$, is achieved by finding the solution of the Equation (1) for $\alpha_g$, thus,

$$\frac{\partial (\rho_g \alpha_g)}{\partial t} + \nabla \cdot \rho_g \alpha_g \vec{v} = 0. \tag{5}$$

Therefore, from the aforementioned considerations, the volume fraction of $\alpha_w$ is computed from the relation $\alpha_w + \alpha_g = 0$.

Because the resultant velocity field is shared by all phases, just one momentum equation is solved for the whole computational domain, which is determined by the volume fractions of all phases through $\rho$ and $\mu$.

$$\frac{\partial}{\partial t} \left( \rho \vec{v} \right) + \nabla \cdot \left( \rho \vec{v} \vec{v} \right) = -\nabla p + \nabla \cdot [\mu (\nabla \vec{v} + \nabla \vec{v}^T)] + \rho g + F, \tag{6}$$

where, $p$, $\mu$, $g$ and $F$ are the density, velocity, pressure in the flow field, viscosity, acceleration due to gravity and the body force respectively. On the other hand, $p$ and $\mu$ are estimated by using, $\rho = \sum_1^p \rho_q \alpha_q$ and $\mu = \sum_1^p \mu_q \alpha_q$.

### 3.5. Interfacial and Surface Tension Treatment

The VOF approach constructs the interface between the fluids using a piecewise-linear method. The interface between two fluids is considered to have a linear slope within each cell. The system uses this linear form to compute fluid advection via the cell faces. The initial stage of interface reconstruction uses the volume fraction and derivatives values in each cell to determine the location of the linear interface respect to the centre of each partially filled cell. The fluid advection across each face is then calculated using the computed linear interface approximation and data about the normal and tangential velocity profile on the face. Finally, the volume flux in each cell is computed using the balance of fluxes obtained in the previous stage.

In addition, the VOF method takes into consideration surface tension at the phase contact. The contact angle between the phases and the wall is specified in the model, and the surface tension coefficient is assumed to be constant. The surface tension model use the continuous surface force model [46] to do this. When surface tension is taken into account in the VOF calculation, a source term, $F$, appears in the momentum equation at which the pressure drop across the surface can be calculated using the surface tension coefficient, $\sigma$, and the curvature of the surface can be estimated using the Young-Laplace equation and two radii in orthogonal directions $R1$ and $R2$, defined as $P_2 - P_1 = \sigma(1/R_1 + 1/R_2)$. As a result, the pressure drop across the surface may be used to describe surface tension. The source term for two-phase water-oil is then written as:

$$F = \sigma \kappa \frac{\rho \nabla \alpha_g}{\frac{1}{2}\left(\rho_g - \rho_w\right)}. \tag{7}$$

The interface curvature $\kappa$ is defined in terms of the divergence of the unit normal, $\hat{n}$, as: $\kappa = \nabla \cdot \hat{n}$, where $\hat{n} = \mathbf{n}/|n|$. Here the surface normal is $\mathbf{n} = \nabla \alpha_g$. The surface curvature is calculated based on the local gradient of the vector normal to the interface, defined as the gradient of the volume fraction of oil $\alpha_g$.

### 3.6. Turbulence Model

If one or more of the phases are in the turbulent regime, turbulence models must be included in numerical simulations of two-phase flow. In this work, the $k - \omega$ Shear Stress Transport (SST) turbulence model [30], which is based on the Reynolds Averaged Navier-Stokes (RANS) technique, was used since it has certain benefits when it comes to high viscosity water-oil flow [47]. Specifically, it is used to make it easier to estimate the onset and degree of flow separation under adverse pressure gradients by including transport effects into the eddy-viscosity approximation over $k - \varepsilon$ turbulence models. In comparison to the $k - \varepsilon$ model it also provides certain advantages in free-stream independence. The model is given by,

$$\frac{\partial}{\partial t}(\rho \kappa) + \frac{\partial}{\partial x_i}(\rho \kappa v_i) = \frac{\partial}{\partial x_j}\left(\Gamma_\kappa \frac{\partial \kappa}{\partial x_i}\right) + \widetilde{G}_\kappa - Y_\kappa + S_\kappa, \tag{8}$$

$$\frac{\partial}{\partial t}(\rho \omega) + \frac{\partial}{\partial x_i}(\rho \omega v_i) = \frac{\partial}{\partial x_j}\left(\Gamma_\omega \frac{\partial \omega}{\partial x_j}\right) + G_\omega - Y_\omega + D_\omega + S_\omega. \tag{9}$$

The term $\widetilde{G}_\kappa$ represents the production of turbulence kinetic energy due to mean velocity gradients; $G_\omega$ the generation of $\omega$: $\Gamma_\kappa$ and $\Gamma_\omega$ the effective diffusivity of $\kappa$ and $\omega$, respectively; $Y_\kappa$ and $Y_\omega$ the dissipation of $\kappa$ and $\omega$ due to turbulence, respectively; $D_\omega$ the cross-diffusion term; $S_\kappa$ and $S_\omega$ are user-defined source terms. The effective diffusivities for $k - \omega$ are the same as in standard $k - \omega$ model.

### 3.7. Sensitivity Analysis and Validation

Four mesh variants were created in order to get numerical results that were not reliant on the mesh. The variations of the outcomes for each mesh version are shown in Figure 3. The data for the sensitivity analysis were collected from the data retrieved using a central line or central marker of the pressure gradient along the calculation domain.

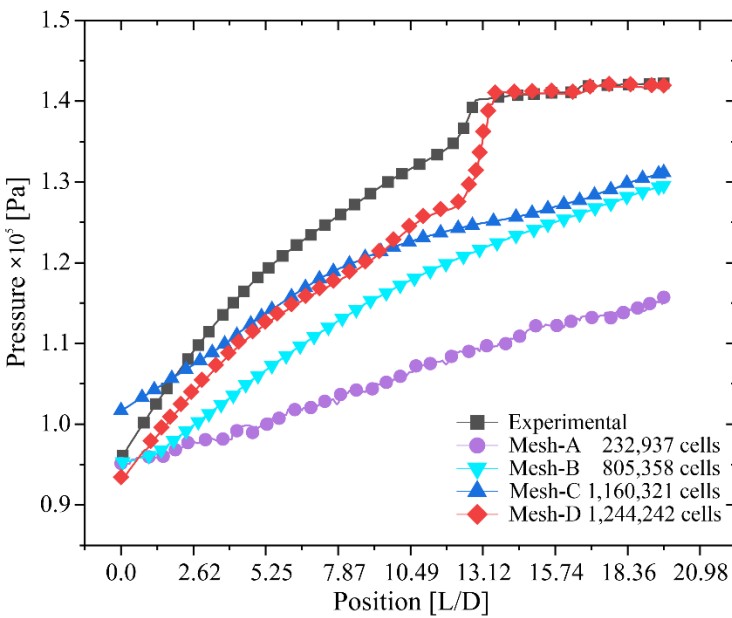

**Figure 3.** Pressure plot, values along the central marker. Experimental data obtained from [33].

Mesh-A significantly underestimates pressure values, suggesting that the representation of flow, i.e., hydrodynamics, was accurately simulated, but the precision of the results is questionable when compared to Meshes -B, -C, and -D. Mesh-B, on the other hand, continues to underestimate pressure values when compared to real data. Despite the fact that the number of cells is slightly over three times that of Mesh-A, 44% less than Mesh-C, and 54% less than Mesh-D, the outcomes are still far from those obtained in the experimental data. Mesh C, for example, displays a region with values that are already near to the experimental ones, but with a 22% of variance, which is still unsatisfactory owing to the needed accuracy of the experimental findings. Finally, Mesh-D exhibits a 9.98% discrepancy from the experimental values essentially in the area of the injectors, but as the flow develops, it stabilizes with a 0.23% deviation before 11.8 $L/D$. More refined meshes beyond this cells number of Mesh-D are not possible owing to computing resources, which might result in unattainable time convergence, forcing the use of LES refined meshes, which need a different kind of model and discretization. Therefore, Mesh-D was selected since the highest precision of the findings is required, as well as the fastest possible convergence time, to be able to characterize the development of the main flow in the early emulsification procedure. For these types of mixing simulations, equilibrium, or stability, is reached with a residual convergence of $1 \times 10^{-5}$ in 15 s of flow. For this reason and mostly due to the confluence, the development of the flow pattern is not yet constituted.

It is worth noting that the deviations from estimated behavior between meshes are ascribed to modelling considerations including the inherent irreversibilities within the process; however, these are deemed not significant based on the strict parametrization conducted and thorough meshing procedure, selection of the discretization schemes, the functions for the treatment of the phases, as well as the boundary conditions. Since the development of the flow pattern is not yet constituted in the initial stage of the pipeline, fluctuations in the results are expected, resulting in these deviations. However, it should be noted that these are not major deviations but are within the parameters for the simulations.

## 4. Results and Analysis

Water is considered the transport phase or continuous phase in this water-oil two-phase research, whereas oil is the dispersed phase. The variables of the continuous phase inside the VOF model will determine the fluid's develop assessment. Having stated so, the calculation and data acquired from the behavior throughout the continuous phase's calculation domain will be used to analyze the evolution of the water-oil flow development. The results will be analyzed by means of slices on XY-plane along the *Z* axis.

The Figure 4 illustrates the phases confluence in the numerical domain. The evolution of the main fluid stands out from case to case in particular, on how the interaction of the phases develops. In this figure, it can be seen that for the cases of double water injection with apparent lateral thrust, in specific Cases 1 and 3, an envelope of the continuous phase is presented over the dispersed phase. That is, when the injection of water supplied by the main pipeline and the lateral nozzle, either right or left (1, 2 or 2, 3), during the first moments $3.93 < L/D < 15.72$ the water surrounds the oil. This observation is not unknown as it bears similarity to the wavy stratified regime accordingly to Angeli and Hewitt [11]. This result entails characteristics for the possible development of narrow emulsions [6,48]. Additionally, a slight turn of the main flow can be seen in the opposite direction of the injection nozzle.

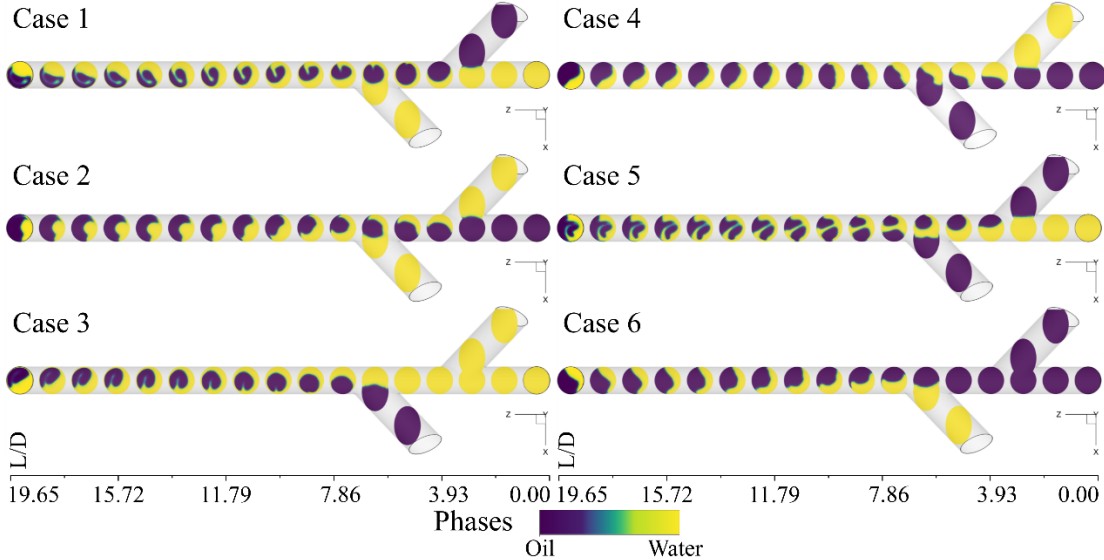

**Figure 4.** Water-oil and oil-water flows development by means of phase fraction.

In an initial assessment, it can be said that there is an appearance of drops at the water oil interface coincident with the interface wavy development, and a possibly narrow emulsion, which could be attributed to the interfacial velocity [6,49]. However, because of the numerical domain total length selected, at the first instance, to study the confluence in a Y-junction, caution should be considered on the analysis of the interface because the development of a specific or complete interface is still evolving.

The lateral confluence in which the phases of Case 2 develop, stabilizes the main flow and confines it towards the centre of the pipe from 11.79 $L/D$ until the exit boundary. In contrast, it resembles a developing annular flow. However, according to the relationship of surface velocities of the flow pattern map, this flow pattern developed at velocities greater than 10 m/s approximately. In this case, a major development and behaviour of the wavy stratified flow could be achieved in lengths greater than 19.65 $L/D$.

When the oil confluence is lateral from 1.96 $L/D$ or 3.93 $L/D$ respectively, the development of the main fluid does not differ much between Cases 4 and 6, but rather by the fluid load inclined towards one side of the main pipeline and opposite to the injection boundary. Still, this behaviour is similar to the wavy stratified flow pattern as in every case as noticed

by Angeli and Hewitt [11]. Main fluid development must continue in order to achieve the real or complete flow pattern and thereby determine if the emulsion generated is narrow or medium [6,11,48].

The Case 5 is the most complicated development achieved within this numerical simulation because it encloses all the characteristics of the cases aforementioned. First, during the confluence, the dispersed phase slightly surrounds the continuous phase, from 3.93 $L/D$ to 5.89 $L/D$. Immediately after, the oil flow comes from the left side injector (3), destabilizing the flow and causing a second zone of envelopment from 5.89 $L/D$ to 13.7 $L/D$. As the flow continues to evolve, the flow pattern could be considered as wavy stratified with the possibility of generating a narrow emulsion [6,48]. However, the interface that develops is not completely defined, at least it is not appreciated through the slices used so far. For all of the above, it is necessary to analyse the velocity contours and corroborate that the interfacial velocity effectively describes the aforementioned flow pattern.

In Figure 5 the oil-water flow development by means of velocity contours of XY multiple slices along the Z axis is depicted. In this analysis, the main fluid velocity contours cannot show the interface or the interface velocity values. However, it is possible to show the velocity values where the phases are located.

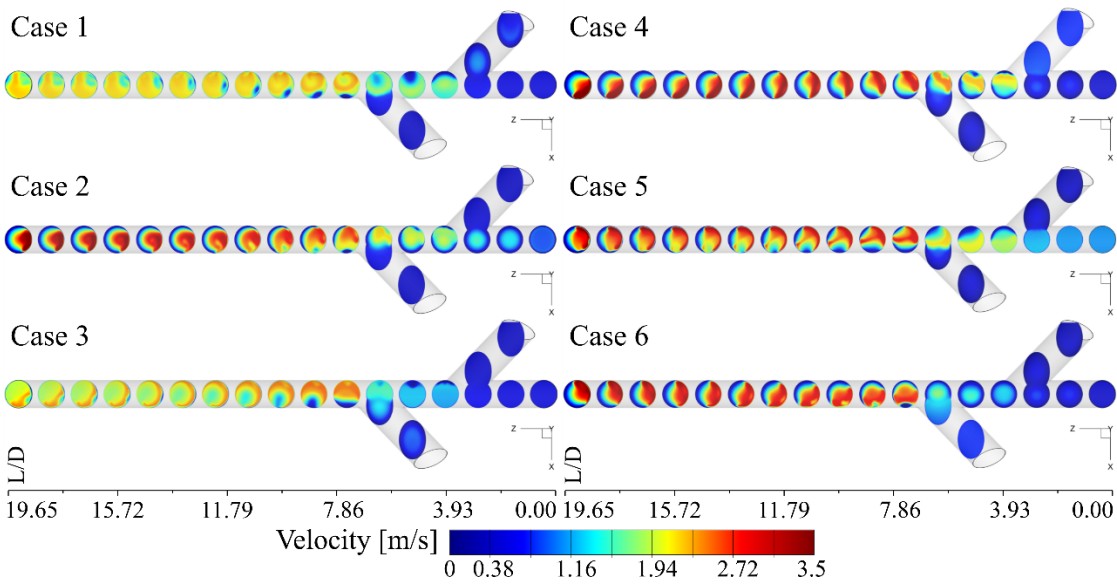

**Figure 5.** Oil-water flow development: velocity contours of multiple slices along the *Z* axis.

In Case 1, as same in Case 3, after the confluence junction $L/D = 6.88$ namely injector 3, the velocity contour in every slice is between $v \approx 1.55$ and $\approx 2.33$ m/s and hardly recognisable phase velocity indicating that phases are moving together with almost the same velocity. Despite that, an interface velocity analysis must be performed in order to determine the phases velocity apart from each other. When water is supply by the later injectors it thrusts the oil performing a subtle blend, which, in many cases, the resulting emulsion is medium [6,11,48].

In Cases 2, 4 and 6, the phases velocities are well defined. For each case, the oil that is moving slower, approximately $v_g \approx 1.6$ m/s, than the water $v_w \approx 2.33$ m/s precipitates to the bottom of the pipe forming a basin as explained Hasson et al. [50]. Specifically, for Case 2 oil-water, and for the other two Cases, of water-oil. Additionally, Trallero et al. [6] and Al-Hadhrami et al. [49] describe how surface velocity determines the flow pattern, and the development of some type of emulsion, be it narrow, medium or wide. They detail that, under a low water-oil surface velocity ratio, the flow is gravity dominated and the phases are totally segregated, where the interface is smooth. This fluid characteristic is particularly of the stratified wavy flow pattern. However, the wavy behavior is normal to the lateral injectors and in the radial direction, and not according to the flow pater map that is the

axial direction normal to the main pipeline inlet boundary. In this specific study, the fluid flow development resembles more the annular flow pattern formation described by Hasson et al. [50] with the possibility of developing a medium to wide emulsion [6,11,48]. In Case 5, the development behavior is rather different, specifically, from 5.89 $L/D$ until 15.72 $L/D$ where the confluence states a different and chaotic blend.

Now, when analyzing the velocity of the interface, information is obtained about the development and evolution of the flow pattern that must be formed under this confluence configuration or Y-junction supply system proposed. Figure 6 presents the complete interface of the water-oil flow development which was obtained by means of iso-surface where water has a 50% mass fraction and 50% of oil mass fraction [51] like in Figure 4. Additionally, three perspective views were selected to facilitate the complete observation of the development of the interface: Isometric, YZ plane and XZ plane of the entire internal computational domain. On the other hand, the velocity contours were placed over the iso-surface to perceive the phases' interaction over this thin layer.

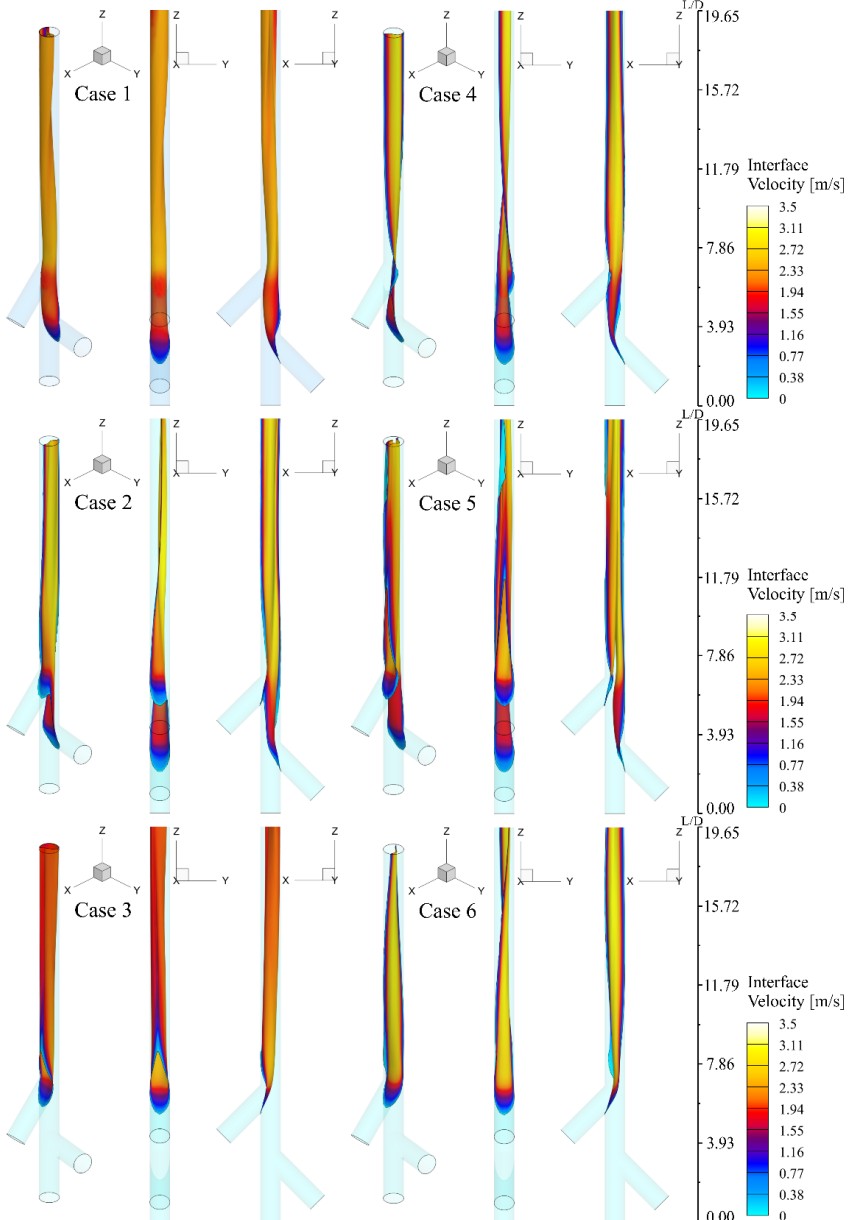

**Figure 6.** Iso-surface of the water-oil interface with the respectively velocity contours with different viewing planes: Isometric, YZ and XZ planes.

In Case 1, the images show that the path of this interface follows, has a *S* shape, starting from 5.89 $L/D$ until the outlet boundary (XZ plane). On the other hand, in the confluence zone, specifically at 5.89 $L/D$, a slightly thrust is exerted on the main fluid where the interface curves (YZ plane). As the fluid evolution continues, a wave that ranges from 7.86 $L/D$ to 15.72 $L/D$ confirms the wavy stratified pattern. At 12.78 $L/D$ the aforementioned oil-water basin is better perceived (isometric). All this trajectory development reveals a delicate twist that the iso-surface undergoes, indicating that a fluid swirl is caused by the lateral confluence thrust. And finally, the velocity contours indicate that both phases move along the numerical domain in almost constant velocity of $v \approx 2.33$ m/s.

In the same way, the Case 2 show the thrusting zone but in a subtle way than Case 1. Contrary to Case 1, the interface fulfils the numerical domain reaching the wall's edge (XZ plane) immediately. The S-shaped path is less noticeable, which better explains the stability of the main fluid such that mixing occurs at the centre of the pipeline. The wavy behaviour starts at $5.89L/D$ but unlike the previous case, the oil remains in the lower zone of the pipe (YZ plane). Here, there are no traces of the slight swirl of the main fluid as in the previous case, although it also still shows the basin that is produced from oil-water. The velocity contours show an increase in the interface velocity, which reaffirms a velocity of the phases $v < 3$ m/s. The water that is located in the upper part moves faster than the oil located in the lower area of the pipe. Despite this, there is an impulse that transmits the water to the oil originating the stratified wavy flow pattern [11], and probably a narrow emulsion [6].

The development and evolution of the conjunction in Case 3 is very similar to what happens in Case 1. However, in this Case, the fluid load hangs towards the right side of the pipe, also causing the main fluid to swirl. While for Case 1 the swirl does clockwise, in Case 3 does counter clockwise. The fluid path is mirror-like S-shape, 7.86 $L/D$ (XZ plane) because the supply system is very much like the opposite of Case 1. The wavy behaviour starts at $11.79L/D$ and the water surround oil without presenting the basin of the previous two cases. The interphase velocity contours show that velocity between phases is slightly lower 1.55 m/s $< v <$ 1.94 m/s.

The Case 4 and 6, has a similar development an evolution than Case 2 with two main differences. One is that the swirl is higher perceptible, counter clockwise at $\approx 10.8$ $L/D$ and clockwise $\approx 9.82$ $L/D$ respectively and two; the surrounding basin is water-oil like noticed Nädler and Mewes [48]. These Cases also present an interface velocity of $v < 3$ m/s.

Despite all these cases' apparently stable water-oil conjunction flow development and evolution, Case 5 is rather chaotic. Two iso-surfaces were created, which ensemble represents the oil-water interphase, specifically meaning that oil surrounding water and an annular flow pattern might be developing [50]. However, there is no conclusive information about what is happening with the water-oil and/or vice versa emulsification, or the specific flow pattern development, at least not according to data described by Nädler and Mewes [48]. In spite of this, it can be ensured that due to the chaotic development and the proportions of the phases, a wide emulsion begins to develop [6,49].

Figure 7 shows the velocity scatterplots of the interface which illustrate the velocity of the main fluid mixture development. By contrasting the results obtained in this work against the flow pattern maps, the early stages of emulsification are observed. The distribution of the oil volume fraction in a cross section in the pipeline for the mixing velocity (main fluid flow) $v > 1.7$ m/s, and the oil and water inlet volume fraction at 50% respectively, describes a concave shape, which in this work is described as basin, behaviour and development also described by Angeli and Hewitt [11]. This occurs in almost all cases with the exception of Case 3 and 5 in which the shape of the interface is complex.

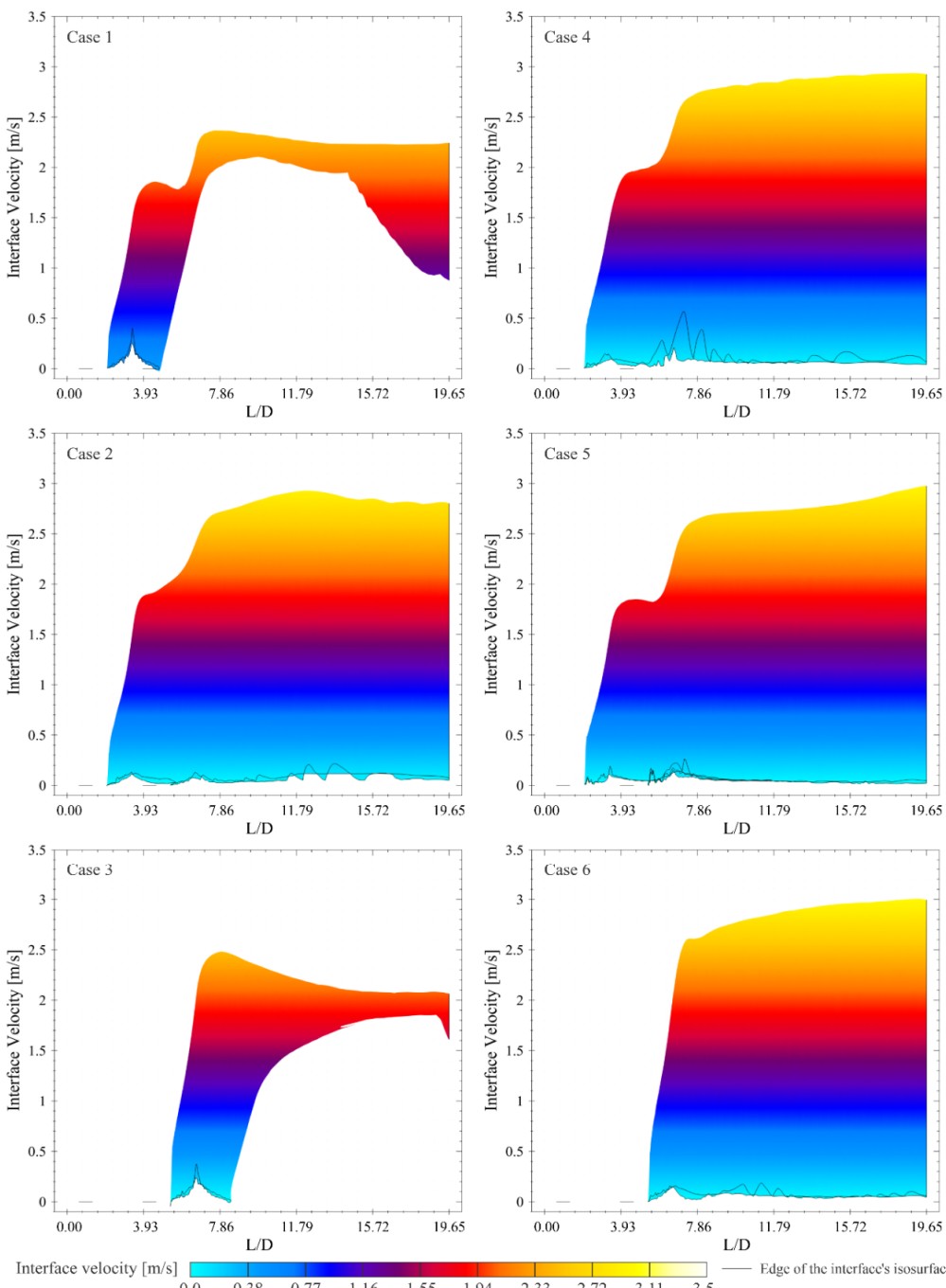

**Figure 7.** Interface velocity scatterplots. The iso-surface that represents the interface is projected over a 2d scatterplot.

For $v > 1$ m/s, it starts the wavy stratified flow pattern and narrow emulsion developing. For $v = 1.5$ m/s, there is no slug flow transition, while for high $v = 2$ m/s, the smooth dispersed droplet flow pattern starts with a narrow to medium emulsion growth. This behaviour is achieved by Cases 1 and 3. However, 4 Cases have $v > 2.5$ m/s which implies that according to Trallero et al. [6], Angeli & Hewitt [11] and Vielma [51] the annular flow pattern is developing. In this way, it is confirmed that the basin formed is due to the development of an annular flow, specifically case 2, 4, and 6 and medium to wide emulsions might be starting to form. For case 5, highly prone to developing wide emulsions [6,11,48].

## 5. Conclusions

Different injection configurations were numerically analysed in order to describe the evolution and development of the water-oil flow pattern in this study. In a horizontal Y-junction pipeline, six distinct phase supply arrangements were described. Through the analysis of the fraction of the phases, as well as the slip velocities, different behaviours were observed. For this reason, the results demonstrate meaningful differences in pattern flow development, attributable mostly to the phase loaded to one or the other side of the pipeline.

Cases 2 and 5 indicate a more complex structure due to relative restriction to the pipeline centre, which is caused by the phases' supply arrangement. The blending process in Case 3 is the smoothest because the continuous phases gradually direct the primary fluid to the pipeline centre. This behaviour is followed by Case 1. The rest of the cases, have a similar development mainly because the high viscous fluid was precipitated to the pipeline bottom modifying the early stage of the flow pattern formation and evolution.

The interfaces velocities confirm the mixture process and the flow pattern development which is driven by the phases' velocities. This is determined precisely when velocities are $1\,\mathrm{m/s} < v < 3\,\mathrm{m/s}$ describing the early stages of the three flow patterns formed during the fluids' confluence obtained with these numerical simulations. In addition, a slight swirl was detected directly attributable to the staggered Y-junction supply system.

Therefore, the supply configuration has a significant relevance on the development of the main fluid flow and substantial extent on the emulsification and the fluid flow pattern development. Finally, care must exercise during the supply system in a Y-junction pipeline to achieve better and smooth blend turning the emulsification process in order to obtain either narrow, medium or wide emulsion.

**Author Contributions:** Conceptualization, M.D.l.C.-Á.; methodology, M.D.l.C.-Á.; software, M.D.l.C.-Á. and I.C.-M.; validation, M.D.l.C.-Á. and L.D.G.S.; formal analysis, M.D.l.C.-Á.; investigation, M.D.l.C.-Á. and I.C.-M.; resources M.D.l.C.-Á.; data curation, M.D.l.C.-Á.; writing—original draft preparation, M.D.l.C.-Á.; writing—review and editing, M.D.l.C.-Á., I.C.-M. and L.D.G.S.; visualization, M.D.l.C.-Á.; supervision, I.C.-M. and J.K.; project administration, J.K.; funding acquisition, J.K. All authors have read and agreed to the published version of the manuscript.

**Funding:** This research work described in this paper was funded by European Union's Horizon 2020 Programme under the ENERXICO Project (Grant Agreement No. 828947) and under the Mexican CONACYT-SENER-Hidrocarburos (Grant Agreement No. B-S-69926).

**Data Availability Statement:** Not applicable.

**Acknowledgments:** The authors of this paper are particularly grateful to Laboratory of Applied Thermal and Hydraulic Engineering (LABINTHAP) from the National Polytechnic Institute of Mexico for the contributions made and the computational resources provided, in collaboration with the Engineering Institute of the National Autonomous University of Mexico, IINGEN-UNAM.

**Conflicts of Interest:** The authors declare no conflict of interest. The funders had no role in the design of the study; in the collection, analyses, or interpretation of data; in the writing of the manuscript, or in the decision to publish the results.

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
