# Peer review of "Numerical Study of Water-Oil Two-Phase Flow Evolution in a Y-Junction Horizontal Pipeline"

_water, doi:10.3390/w14213451_

Round 1

Reviewer 1 Report

1. What is the Innovativeness in the study?

2. Which model you had study for evaluation?

3. Which control measures you adopted for your study?

4. Have you developed Risk Model? If yes - how and if now - why

5. Limitations in the study?

6. Have you compared your work with other researchers?

7. Any assumptions have you considered in your work ?

Major revision

Author Response

Dear Reviewer, the attached document encloses the answers to your comments. The authors appreciate the thorough revision of the submitted manuscript.

Please see the attachment file

Reviewer 2 Report

L65-67: This sentence is not well understood. Could you rephrase it so that any reader will be confused?

L64-100: The authors have put together this complete section, mixing several contents into one. I think that the part corresponding to the objectives of the work (L65-81) should be included, in the introduction, as a sub-section called "objectives". While the rest of the paragraphs (L82-100) should be accommodated in the section Material and Methods, structured as the authors see fit, but... "outside" the objectives section. The justification of the methods used, which has already been described in the introduction, should be included in Material and Methods. I think this should be corrected. The way it is, it mixes contents of different sections, which leads to confusion.

L103: “Engineering Institute-UNAM”. What does UNAM mean? This is the first time this abbreviation has appeared without its meaning being clear. I have looked at the authors' signature and... it doesn't appear either. I think it would be necessary to name it in full.

L111: In Figure 1, shouldn't "L/D ratio", instead of just "L/D", appear as the horizontal reference scale? And, if it does not correspond to what this reviewer is interpreting, how could the authors make this figure self-explanatory? That is, the Figure caption alone should explain all that the Figure contains. Check it out!

L118: If the authors are using commercial software, which requires learning how to use, they should reference where any reader who wants to reproduce their results should go to carry out the same experiment they have carried out. Therefore, it would be advisable to "cite bibliographically" which are the sources to learn the skills in the use of such software. For example:

-        Tutorial (2015); FLUENT Tutorial guide. ANSYS, Inc. Release: 16.0, January 2015

-        ANSYS Theory (2015) ANSYS FLUENT Theory guide. ANSYS, Inc. Release: 16.2, July 2015

Also, in that way, justice is done to the trademark/team that develops the software.

L132-133: …”it is necessary to discard the chemical dynamics [36,37] and the hydrogen bonding velocity [38] during the conjunction”…

Well, what does that mean? What do the authors mean by "discard"?

Does the chemical dynamics no longer have any effect on the experiment? How do you "discard" the chemical dynamics and the hydrogen bonding rate? Are these conditions made possible by the software? Is this possible in reality?

So many unanswered questions for so little explanation!

L268-291: I think I have overlooked where the authors have explained how: with what parameters, they were going to compare the curves they show in Figure 3. Could you please point this out to me? Perhaps some readers might also be interested in comparing curves with each other!

L294-302: Readers, and this reviewer, must therefore take the authors' statements as true, without reference citations or bibliographic references.... parameter values, which allow us to contrast this "non-significance" or "not major deviations"? Don't the authors think they could be more precise in these "general statements" about the goodness of their "way of proceeding", based on other works on the same subject?

L304-312: Well, these are the authors' "recommendations" on how to proceed, so they should not be included in the results section, but in one of the Material and Methods sections.

L457-459: I think that with the first sentences, the authors want to introduce the conclusions. But... they look like a small summary that should not appear in the conclusions. If they want, they can rephrase it until they get to: "The results demonstrate significant differences…”

Nevertheless: "significant differences"???? We have analysed the paper and nowhere does it specify which statistical instrument the authors have used to make this claim?

Or have I missed it? Could you please clarify?

L473-477: This paragraph is the only one that seems to me to be a "real" conclusion of the work presented here. And yet, while reading the text, it did not seem to me that the authors gave prominence to "in order to obtain either narrow, medium or wide emulsion".

Why is it so important? Perhaps, if it is, it should be worked a little more in the text.

Author Response

Dear Reviewer, the attached document encloses the answers to your comments. The authors appreciate the thorough revision of the submitted manuscript.

Round 2

Reviewer 1 Report

revision is appropriate